# Increased force and elastic energy storage are not the mechanisms that improve jump performance with accentuated eccentric loading during a constrained vertical jump

Eric Yung-Sheng Su[1,2☯]*, Timothy J. Carroll[1☯], Dominic J. Farris[1,3☯], Glen Lichtwark[1,2☯]

**1** School of Human Movement and Nutrition Sciences, The University of Queensland, Brisbane, Queensland, Australia, **2** School of Exercise and Nutrition Sciences, Queensland University of Technology, Kelvin Grove, QLD, Australia, **3** Faculty of Health and Life Sciences, Public Health and Sport Sciences, University of Exeter, Exeter, Devon, United Kingdom

☯ These authors contributed equally to this work.
* yungsheng.su@uq.edu.au

**Editor:** Žiga Kozinc, Faculty of Health Sciences, University of Primorska, SLOVENIA

**Data Availability Statement:** All relevant data are within the manuscript and its Supporting Information files.

## Abstract

Accentuated eccentric loading (AEL) involves higher load applied during the eccentric phase of a stretch-shortening cycle movement, followed by a sudden removal of load before the concentric phase. Previous studies suggest that AEL enhances human countermovement jump performance, however the mechanism is not fully understood. Here we explore whether isolating additional load during the countermovement is sufficient to increase ground reaction force, and hence elastic energy stored, at the start of the upward movement and whether this leads to increased jump height or power generation. We conducted a trunk-constrained vertical jump test on a custom-built device to isolate the effect of additional load while controlling for effects of squat depth, arm swing, and coordination. Twelve healthy, recreationally active adults (7 males, 5 females) performed maximal jumps without AEL, followed by randomised AEL conditions prescribed as a percentage of body mass (10%, 20%, and 30%), before repeating jumps without AEL. No significant changes in vertical ground reaction force at the turning point were observed. High load AEL conditions (20% and 30% body weight) led to slight reductions in jump height, primarily due to decreased hip joint and centre of mass work. AEL conditions did not alter peak or integrated activation levels of the knee extensor muscles. The constrained movement task used here, which excluded potential contributions of trunk motion, arm swing, rate of descent, squat depth, and point of load application, allows the conclusion that increased elastic energy return is not the primary mechanism for potentiating effects of AEL on jump performance.

## Introduction

Accentuated eccentric loading (AEL) is a form of movement manipulation that has been suggested to enhance power output during stretch-shortening cycle exercises. For example, during

**Funding:** Su, E. Y. received University of Queensland Graduate School Scholarship (UQGSS) with tuition fee offset and living allowance stipend during PhD candidature while undertaking this research. Lichtwark, G. was supported by Australian Research Council Future Fellowship (FT190100129) while undertaking this research. None of the funders play any role in the study design, data collection and analysis, decision to publish, or preparation of the manuscript.

**Competing interests:** The authors have declared that no competing interests exist.

a human countermovement jump (CMJ), AEL requires an external load to be added to the body during the downward (eccentric) movement and then released at the transition from the downward to the upward (concentric) movement. Studies have indicated that the immediate response to AEL during a CMJ could increase jump height by ~4.3–9.5%, increase peak power output by ~9.4–23.2%, and increase maximal concentric vertical ground reaction force by ~3.9–6.3% [1,2]. By contrast, Aboodarda et al. [3] reported that AEL applied through elastic resistance in human drop jumps did not alter jump height, muscle activation level, or other kinetic profiles during push-off. There is conflicting evidence on the performance-enhancing effect of AEL CMJ, with studies showing both positive [1,2,4] and no effect [3,5–7]. A review study concluded that current evidence for acute responses to AEL is inconsistent, possibly due to different exercises selected, equipment used, range of loads, or participants' characteristics across different experiments [8].

There are a number of potential mechanisms that might drive enhanced power output during AEL movements. One common explanation for why AEL should enhance power is that increased load in the eccentric phase amplifies elastic energy storage in the tendon and aponeurosis, which can be released in the concentric phase [9]. For instance, AEL CMJ may result in greater force generation in the descent to decelerate the added mass or resist the added force, potentially resulting in greater tendon loading and strain energy storage prior to the upward motion (strain of elastic tissues is directly related to force applied). Alternatively, AEL might change the coordination strategy, allowing muscles to generate greater force or power during the subsequent concentric portion of the jump. This might be achieved through increased squat depth [10,11] or changes in the position of the centre of mass (COM) relative to the joints [12]. Whilst there is some evidence that humans can achieve greater jump heights through AEL without significant changes in squat depth [2], a combination of the factors above might contribute to enhanced jump performance in AEL.

Here we sought to investigate whether isolating the effect of additional load (in the absense of other potential contributors) during an AEL CMJ can increase the force applied to the ground at the start of the concentric movement in a CMJ and enhance positive power generation during the jump. We used a constrained AEL scenario to control multiple alternative factors that might influence force and power. Specifically, we controlled squat depth, fore-aft and lateral movement of the trunk, arm swing and the point of application of the added mass during the CMJ. Without these constraints, it would be impossible to identify what aspects of AEL might enhance performance. For instance, altered kinematics could also affect performance outcomes of a jump. Here we we address the specific question of whether there is enhanced elastic storage and return of energy in the hip, knee and ankle extensors purely due to added load. We hypothesised that under these controlled conditions, AEL would increase the vertical ground reaction force (VGRF) at the start of the concentric movement and therefore increase the power and subsequent work during the concentric phase of the CMJ. We also hypothesised that the increased force would require greater muscle activity of the lower limb muscles undergoing stretch-shortening cycles, particularly the knee extensor muscles that are largely responsible for power production during the jump.

## Materials and methods

### Sample size calculation

We conducted a pilot study with the similar experimental setup as Sheppard et al. [2] From our pilot study, we found an improvement in jump height with AEL with an effect size of 0.83 (paired analysis). Our sample size calculation found that 11 participants were sufficient to achieve a statistical power of 80% (within-factor effect size = 0.83, alpha = 0.05, one-tailed).

We recruited one additional participant to account for potential data loss or improve the statistical power.

## Participant characteristics

Twelve healthy and recreationally active adults (7 males, 5 females, height = 177 ± 8.1 cm, mass = 75.3 ± 10.7 kg, age = 28.1 ± 6.7 years) gave written informed consent to participate in this study. Ethical approval was granted from the institutional ethics review committee at The University of Queensland (approval number: 2021/HE001129). The recruitment period for this study was from 15/10/2021 to 10/2/2022. Our study required participants to achieve an effective jump height of at least 40 cm (male) or 30 cm (female) using a vertical jump-and-reach device (Swift Performance, Wacol, QLD, Australia). The effective jump height definition was consistent with the Exercise and Sports Science Australia testing protocol [13]. Participants were screened for their jump ability prior to participation.

## Testing protocol

Participants attended two laboratory sessions, one for familiarization (session one) and the other for data collection (session two). In session one, participants performed 20–30 practice jumps on a jumping sled (see below–Jumping Sled) with and without AEL from various squat depths, with at least 1 minute rest between jumps. The sled only allowed the trunk to translate in the vertical direction, with no trunk rotation or fore-aft or lateral translation. Participants practiced non-AEL and AEL jumps with 10%, 20%, and 30% additional body mass attached to the sled. The non-AEL condition included the mass of the backrest, ~7 kg, and the other AEL conditions included extra mass in addition to the mass of the backrest.

Participants attempted maximal effort jumps with both hands holding onto bars located on the backrest above the shoulders, thus preventing arm swing. Participants were verbally motivated with the instruction to "jump as high as possible" after each maximal trial. Based on the jumping performance in the non-AEL condition, we selected an appropriate squat depth for each participant that maximised jump height, which was used as the controlled squat depth for the remainder of the study. Participants practiced CMJ from this same depth across different AEL conditions. Squat depth was monitored live and recorded based on a string potentiometer attached to the sled. Visual feedbacks of squat depth and jump height were given after each jump.

In session two, we collected force plate, motion capture, and electromyography (EMG) data to examine jump performance under different AEL conditions. Participants performed 3~5 submaximal non-AEL jumps to warm-up. They then performed at least 3 valid maximal jumps at their controlled squat depth (recorded in session one) for each of the different conditions. Participants were required to achieve the specified squat depth to qualify for a valid trial, with a maximum of 5 cm error margin below this depth. This was monitored using the string potentiometer with visual feedback given after each trial. Participants performed maximal jumps with non-AEL condition ($BW_{pre}$) first, followed by AEL conditions (10%, 20%, and 30% AEL) in a randomised order, and finished with another non-AEL condition ($BW_{post}$). At least 1 minute rest was provided between each maximal trial.

## Jumping sled

The jumping sled design is shown in Fig 1. An aluminium backrest (sled) was attached to a squat cage fitted with linear rails and bearings. The sled included an electromagnet, shoulder bars, and a waist belt for participants. A pulley system with an inextensible metal wire (diameter: 5 mm, marine grade stainless steel cable) and weight plates was used to add or remove

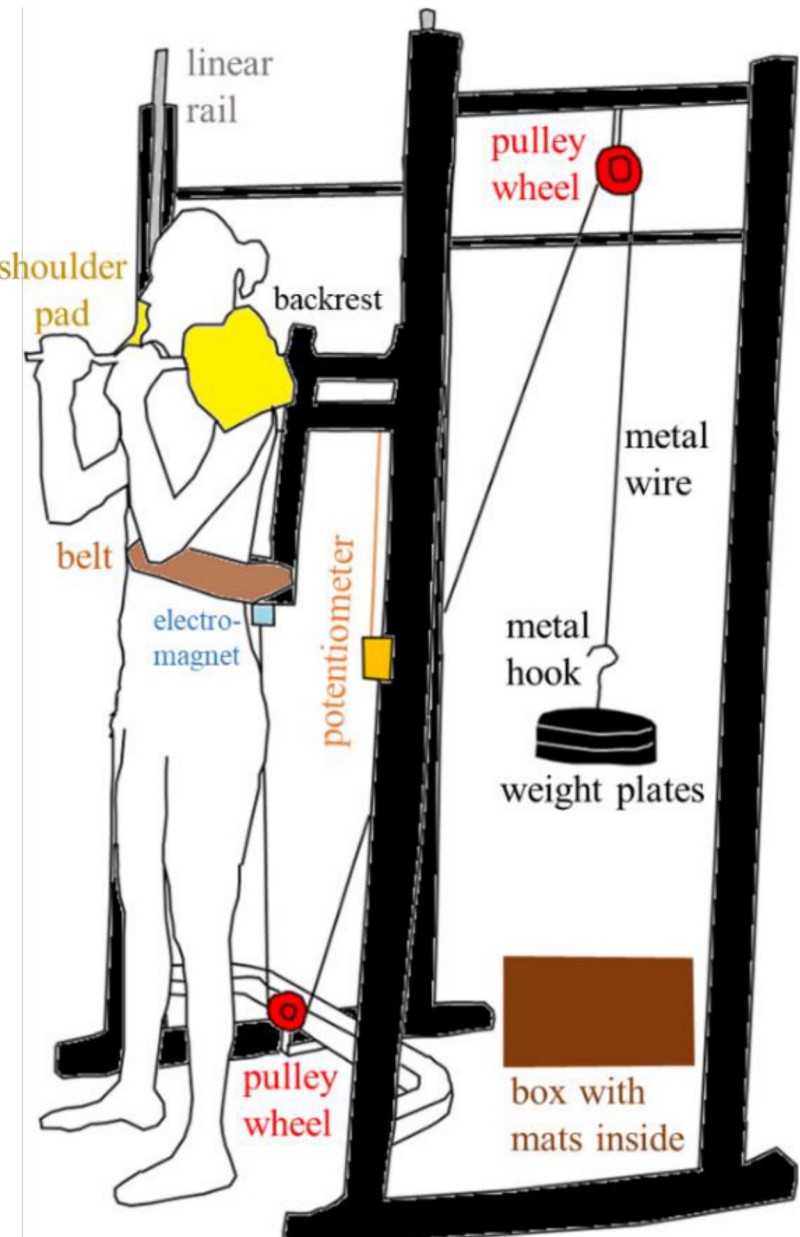

**Fig 1. Jumping sled design.** The squat cage (black) was fitted with linear rails (grey) and an aluminium backrest (black). An electromagnet (blue) was attached to the bottom of the backrest, and two shoulder bars with foam padding (yellow) were fixed at the top of the backrest. Participants were strapped firmly on the backrest with a waist belt (brown). Two pulley wheels (red) were fixed on the squat cage, and a metal wire was used to carry a metal hook and weight plates (black). A linear string potentiometer (orange) was attached between the fixed frame and the sliding backrest. The weight plates were released into a box (dark brown) behind the participant.

extra mass. The friction in the linear rail system and pulleys was not considered in our analysis but was assumed to be negligible. One end of the wire was attached to a light iron disc that would hold to the electromagnet when charged. A linear string potentiometer (SP2-50, TE Connectivity, Berwyn, PA, USA) was fixed on the squat cage to measure the sled's vertical position. An electromagnetic switch was used to release the added mass when the participant reached the lowest depth during the jump. The mass was dropped safely into a box behind the

participant. Since participants were explained the protocol and were trained with the noise of drop weights hitting the box in the familiarisation session (session one), we believe that participants were aware of the noise and were unlikely to be hugely affected by the noise during jumping in the data collection (session two). We ensured that the weight plates were static prior to the start of the trial, which ensured that the downward movement of the mass did not incur additional horizontal movements that might contribute to added forces in the cable through centripetal accelerations.

We recorded the calibrated potentiometer length in the A/D board (Micro 1401, Cambridge Electronic Device, Cambridge, United Kingdom) and ran a custom script to determine the release timing in the Spike 2 software (Cambridge Electronic Device, Cambridge, United Kingdom). We defined a threshold length equivalent to each participant's controlled squat depth. The system monitored when the potentiometer velocity turned positive after crossing this threshold to ensure the additional mass was only released when the lowest depth was achieved. Once the participant reached the required depth and initiated the push-off phase, the Spike 2 software sent a TTL pulse to deactivate the electromagnet, causing the added mass to drop. On average, there was a delay of 0.084 ± 0.050 seconds from the turning point to TTL generation, during which the average torso COM displacement was 2.81 ± 1.84 cm (or 6.9% of the average upwards jump displacement).

### Data collection

**Motion capture and force plates.** Reflective markers (9 mm) were placed on the participant's skin and tracked with an 11-camera, 3D optoelectronic system (Oqus, Qualisys, AB, Sweden). These markers were placed on the following anatomical landmarks on the trunk and both legs: acromion process, jugular notch, xyphoid process, ASIS, PSIS, iliac crest directly superior to the greater trochanter, medial and lateral joint centre of the knee, medial and lateral malleoli, sustentaculum tali (medial calcaneus marker), fibular trochlea (lateral calcaneus marker), calcaneal tuberosity (posterior calcaneus marker), first and fifth metatarsal, interphalangeal joint of the big toe. A cluster of 4 markers were also attached to both the shanks and thighs to track motion. We also added one marker on each side of the deltoid muscle belly to allow tracking of the trunk segment motion. Marker positional data were sampled at 125 Hz and collected in Qualisys Track Manager (QTM) software (Qualisys, Gothenburg, Sweden). Two tri-axial AMTI in-ground force plates (AMTI, Watertown, MA, USA) synchronously collected ground reaction force (GRF) data at 1250 Hz in QTM. Prior to jumping trials, we collected a static trial for each participant, where they stood with legs evenly spaced on two force plates and their hands positioned similarly to the jumping trials.

**Electromyography.** Surface EMG data were collected from six lower limb muscles on the right leg with a wireless EMG system (MYON m320; MYON, Schwarzenberg, Switzerland). Muscles recorded were soleus (SOL), medial gastrocnemius (MG), vastus lateralis (VL), rectus femoris (RFEM), biceps femoris long head (BFEM), and gluteus maximus (GMAX). Bipolar EMG electrodes of 24 mm diameter were placed over the belly of each muscle according to the SENIAM guidelines [14]. EMG data were synchronously recorded with motion capture data in QTM at 1250 Hz and exported as MAT file for further analysis in MATLAB (Mathworks, Natick, MD, USA).

### Data processing and analysis

**Kinematics and kinetics.** Motion capture markers were labelled, tracked, and digitally exported with EMG and GRF data. We used OpenSim [15,16] and a publicly available model [17] that was first scaled to each participant and then performed inverse kinematics and

inverse dynamics calculation in OpenSim. Scaling factors were calculated by dividing participant-specific marker distances by the corresponding distances on the generic model. Inverse kinematics results and experimental GRF data were filtered with a 15 Hz low-pass second-order Butterworth filter prior to inverse dynamic calculation. OpenSim outputs data were then imported into MATLAB for further analysis, however, we only analysed trials that achieved the highest effective jump height in each condition for each participant for kinematic and kinetic data.

In MATLAB, we calculated effective jump height as the difference between the highest vertical position of the manubrium marker (MAN) for each trial and the average MAN vertical position in the static trial for each participant. The start of the jump was defined as the instant when the VGRF decreased by more than 5 N from the initial standing phase for each trial. The turning point of the jump was defined as the lowest vertical position of the MAN for each trial. Take-off was estimated when the right vertical GRF dropped to zero. Push-off phase was defined from the turning point to take-off. Squat depth was defined as the difference between the average MAN vertical position in the static trial for each participant and the lowest MAN vertical position for each trial. Maximal descent speed was determined as the largest magnitude of the MAN negative velocity during the descent. We took the dot product of the model's joint moment and joint angular velocity to calculate joint power for each joint (right hip, knee, ankle) across all degrees of freedom. Positive values in joint moments represent hip extension, knee extension, and ankle plantarflexion internal moments. Vertical COM power during the concentric phase was calculated as the dot product of the model's vertical COM velocity (from OpenSim Body Analysis) and the experimentally collected total vertical GRF data (sum of two vertical GRF from each plate). Joint moments, joint power, and vertical COM power data were filtered with a 5 Hz low-pass second-order Butterworth filter. We then integrated joint power and vertical COM power during the push-off to calculate concentric joint work and concentric vertical COM work. We also integrated negative joint power during the eccentric phase (descent) to calculate eccentric joint work. Finally, we calculated the total lower limb joint work by summing the concentric work across all degrees of freedom at the hip, knee, and ankle joints.

**Electromyography.**    Raw EMG data were zero-phase band-pass filtered (30–350 Hz second-order Butterworth filter), rectified, and zero-phase low-pass filtered (5 Hz second-order Butterworth filter) to form a linear envelop as the processed EMG data. Due to an acquisition error, some trials in the AEL conditions had incomplete raw EMG data that only ranged from the start of the jump until just after the turning point. We excluded trials with incomplete EMG data, and reduced data to two key EMG outcome measures for each participant and condition: the peak EMG amplitude achieved during the entire jump and the integrated EMG during push-off phase. These outcome measures (peak and integrated EMG) were calculated by averaging across multiple trials for each condition per participant. For each participant, the peak EMG amplitude obtained across all trials was used to normalise all EMG values, as per consensus recommendations [18].

**Statistical analysis.**    Linear mixed-effects models with Restricted Maximum Likelihood solution were used to compare AEL conditions (BW$_{pre}$ [Non-AEL], 10%, 20%, 30% AEL). A mixed-effects model to perform repeated measures analysis was used to account for one missing data point in the 30% AEL condition. We reported the p-value and the partial eta squared ($\eta^2_p$) effect size for each main effect. Dunnett's multiple comparison tests were used to detect the origin of significant main effects between experimental conditions (10%, 20%, 30% AEL) against one single control condition (BW$_{pre}$). We reported the p-value and the Cohen's d effect size for each multiple comparison test. Paired t-tests were used to compare non-AEL conditions before (BW$_{pre}$) and after (BW$_{post}$) AEL interventions, and the p-value and the Cohen's d

effect size were reported. Alpha was set at 0.05. Statistical assumptions for each dependent variable were performed using the D'Agostino-Pearson normality test and all data were normally distributed. Statistical analysis was performed using GraphPad Prism 8.

## Results

### Jump height and rate of descent

There was a significant main effect of AEL condition on jump height (p = 0.014, $\eta^2_p$ = 0.020) and rate of descent (p = 0.0004, $\eta^2_p$ = 0.141). Multiple comparison tests found a significant decrease in mean effective jump height for 20% (p = 0.028, d = 0.88) and 30% (p = 0.013, d = 0.85) AEL conditions compared to the baseline (BW$_{pre}$); however, there was no significant difference in effective jump height for 10% (p = 0.146, d = 0.60) AEL condition (Fig 2A). Participants showed mixed individual responses in effective jump height before (BW$_{pre}$) and after (BW$_{post}$) AEL interventions (Fig 2B), such that there were no significant differences in effective jump height between the non-AEL conditions (p = 0.648, d = 0.136). There was a significant decrease in maximal descent speed in all three AEL conditions (10%: p = 0.011, d = 0.491; 20%: p = 0.001, d = 0.774; 30%: p = 0.001, d = 1.130) compared to the baseline (BW$_{pre}$) from the multiple comparisons (Fig 2C).

### Squat depth

There was no significant main effect of AEL condition on squat depth (p = 0.107, $\eta^2_p$ = 0.022). Therefore, we were confident that squat depth control was effectively implemented in this experiment, and there was no systematic bias created by differences in depths across conditions.

### Vertical ground reaction force

The average time-varying VGRF across each condition is shown in Fig 3A. Although VGRF was visually higher with increasing AEL load early in the downward movement, there was no

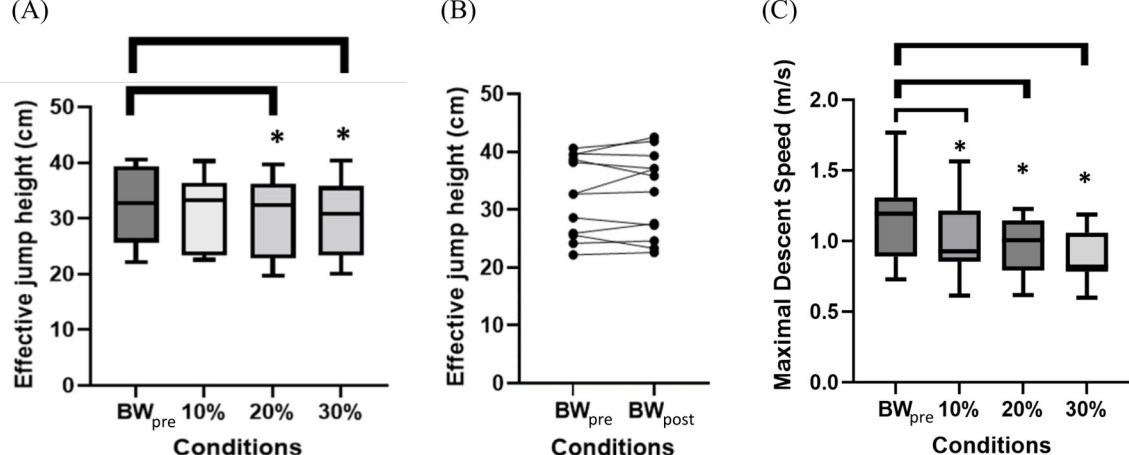

**Fig 2. Results for effective jump height and maximal descent speed.** (A) Box and Whisker plot for effective jump height across non-AEL (BWpre), 10%, 20%, and 30% AEL conditions. The x-axis shows different loading conditions, and the y-axis shows the effective jump height (cm). The asterisks represent a statistically significant difference from the mean of the baseline (BWpre) condition. (B) Effective jump height for non-AEL conditions before (BWpre) and after (BWpost) AEL interventions across individual participants. The x-axis shows different conditions, and the y-axis shows the effective jump height (cm). The black dots represent each selected trial for each participant per condition. (C) Box and Whisker plot for maximal descent speed across non-AEL (BWpre), 10%, 20%, and 30% AEL conditions. The x-axis shows different loading conditions, and the y-axis shows the maximal descent speed (m/s). The asterisks represent a statistically significant difference from the mean of the baseline (BWpre) condition.

(A)                                                            (B)

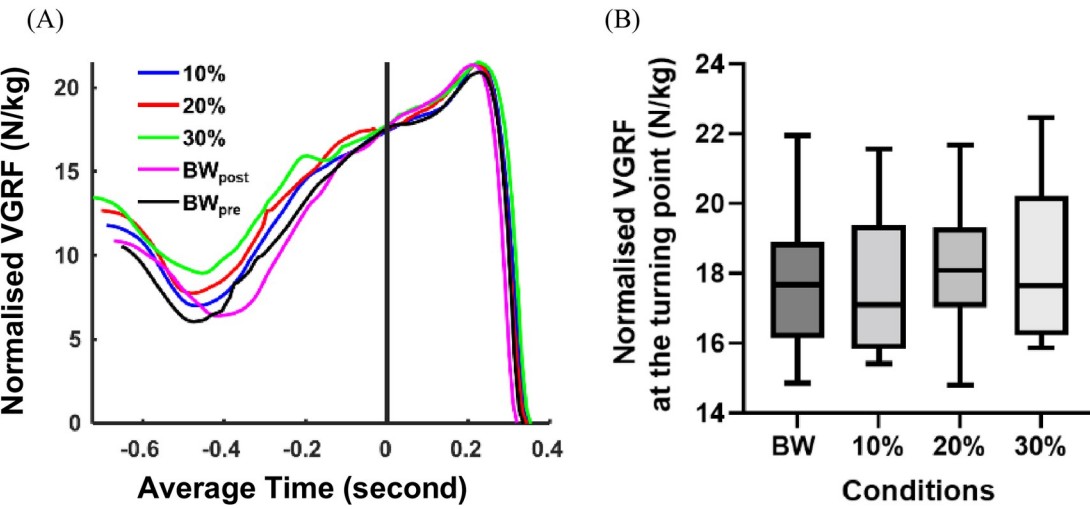

**Fig 3. Results for vertical ground reaction force (VGRF).** (A) Average time varying VGRF across each condition. The x-axis shows the average time (seconds), and the y-axis shows the body mass normalised VGRF (N/kg). Each colour coded line represents the average value across all participants for each condition. Time 0 represents the bottom of the countermovement (turning point). (B) Box and Whisker plot for body mass normalised VGRF at the turning point across non-AEL (BWpre), 10%, 20%, and 30% AEL conditions. The x-axis shows different loading conditions, and the y-axis shows the body mass normalised VGRF (N/kg).

significant difference (p = 0.323, $\eta^2_p$ = 0.024) in the magnitude of VGRF at the bottom of the countermovement (turning point) across AEL conditions (Fig 3B).

## Joint kinetics and energetics

There was a significant main effect of AEL condition on normalised hip extension moments (p = 0.005, $\eta^2_p$ = 0.107) and knee extension moments (p = 0.014, $\eta^2_p$ = 0.049) at the turning point. There was a significant decrease in the magnitude of the normalised hip extension moments at the turning point in all three AEL conditions (10%: p = 0.002, d = 1.337; 20%: p = 0.012, d = 1.024; 30%: p = 0.024, d = 0.808) compared to the baseline (BW_pre) from the multiple comparisons. The normalised knee extension moments at the turning point showed significant increases in 20% (p = 0.005, d = 1.181) and 30% (p = 0.016, d = 1.292) AEL conditions from the multiple comparisons. There were no significant differences across conditions for the normalised ankle plantarflexion moment at the turning point (p = 0.202, $\eta^2_p$ = 0.030) from the mixed-effect model. The average joint moments (across all joints) relative to time for each condition can be found in S1 Fig.

The average time-varying right hip, knee, and ankle joint powers across each condition are shown in Fig 4A. There was a significant main effect of AEL condition on positive peak normalised hip joint power during push-off (p = 0.041, $\eta^2_p$ = 0.059). Multiple comparison tests showed a significant reduction in the positive peak normalised hip joint power during push-off in 10% (p = 0.002, d = 1.371) and 20% (p < 0.0001, d = 1.955) AEL conditions compared to the baseline (BW_pre). However, the mixed-effect model found no significant differences in positive peak normalised knee (p = 0.215, $\eta^2_p$ = 0.007) and ankle (p = 0.661, $\eta^2_p$ = 0.0002) joints power during push-off (Fig 4B). We also found that there was a significant main effect of AEL condition on the peak magnitude of normalised eccentric (negative) hip joint power (p = 0.002, $\eta^2_p$ = 0.171). There was a significant decrease in the peak magnitude of normalised eccentric hip joint power in all three AEL conditions (10%: p = 0.026, d = 0.776; 20%: p = 0.009, d = 0.823; 30%: p = 0.009, d = 0.857) compared to the baseline (BW_pre) from the multiple comparisons.

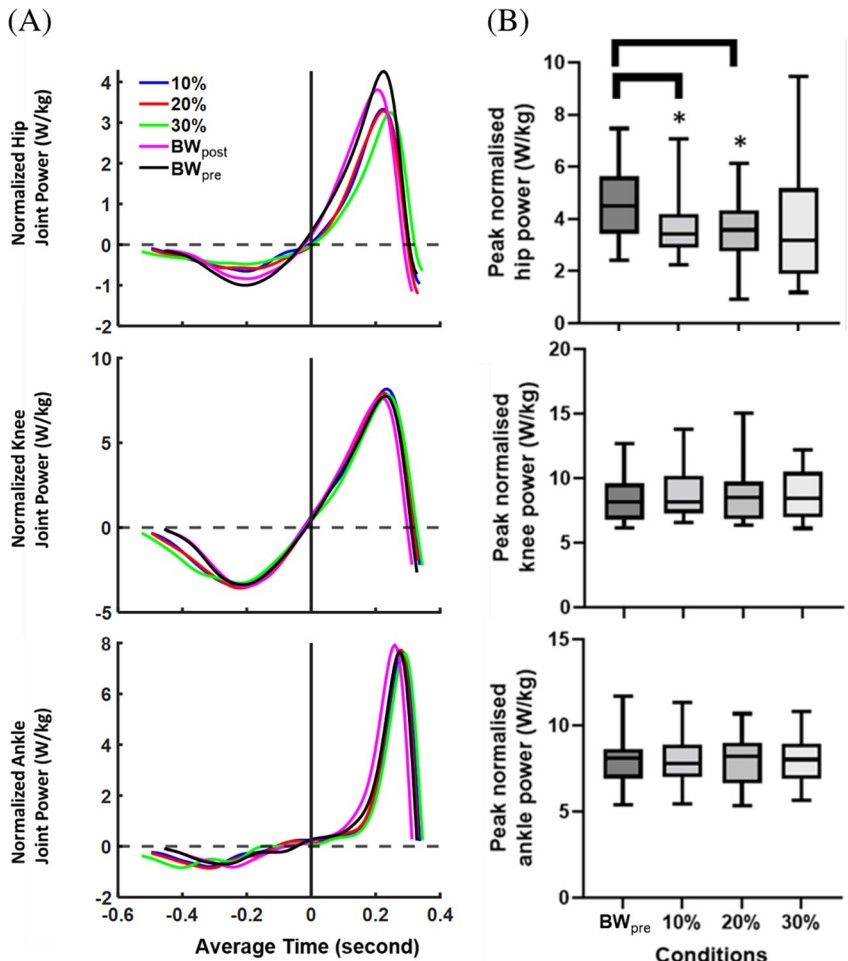

**Fig 4. Results for right lower limb joint power.** (A) Average time varying right hip, knee, and ankle joint power across each condition. The x-axis shows the average time (second), and the y-axis shows the body mass normalised joint power (W/kg). Each colour coded line represents the average value across all participants for each condition. Time 0 represents the bottom of the countermovement (turning point). (B) Box and Whisker plots for peak normalised right hip extension, knee extension, and ankle plantarflexion joint power during push-off across non-AEL (BW_pre), 10%, 20%, and 30% AEL conditions. The x-axis shows different loading conditions, and the y-axis shows the peak normalised hip extension, knee extension, and ankle plantarflexion joint power (W/kg). The asterisks represent a statistically significant difference from the mean of the baseline (BW_pre) condition.

There was a significant main effect of AEL condition on hip joint work ($p = 0.001$, $\eta^2_p = 0.102$) and sum of joints work ($p = 0.016$, $\eta^2_p = 0.023$) during push-off. However, we found no significant main effects in the knee ($p = 0.302$, $\eta^2_p = 0.016$) and ankle ($p = 0.418$, $\eta^2_p = 0.004$) joint work during push-off. Multiple comparison tests showed a significant reduction in the hip joint work during push-off in all three AEL conditions (10%: $p = 0.001$, $d = 1.559$; 20%: $p = 0.001$, $d = 1.550$; 30%: $p = 0.01$, $d = 0.991$) compared to the baseline (BW_pre). We also found a significant reduction in the sum of joints work in all three AEL conditions (10%: $p = 0.005$, $d = 1.174$; 20%: $p = 0.017$, $d = 0.967$; 30%: $p = 0.001$, $d = 1.200$) compared to the baseline. Data for concentric joint work (across all joints) for each condition can be found in S2 Fig. We also showed the change in negative joint work during the eccentric phase across different conditions (S3 Fig). We examined the negative work done at each joint and found significant main effects of AEL condition on negative work done at hip ($p = 0.0003$, $\eta^2_p =$

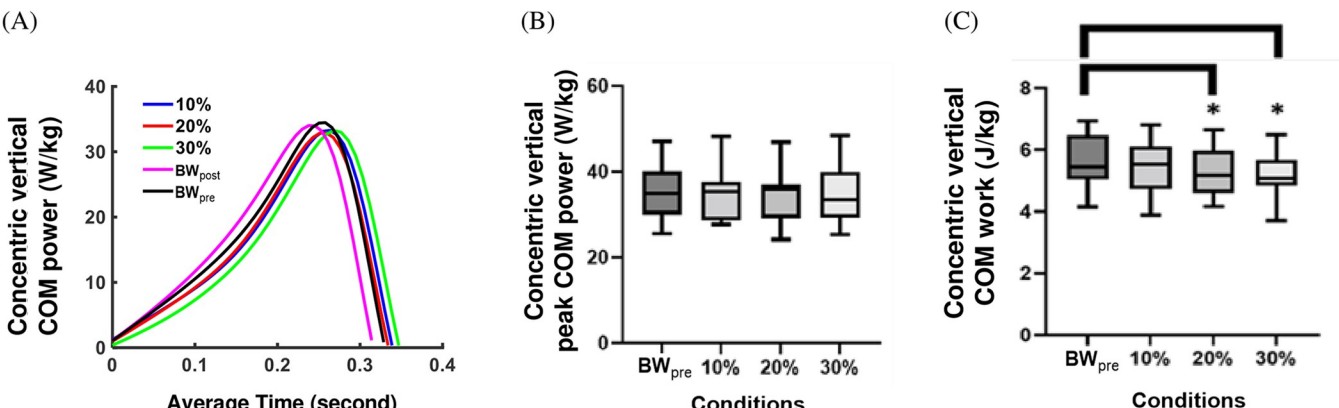

**Fig 5. Results for whole-body vertical centre of mass (COM) power and work.** (A) Average time varying concentric vertical COM power during push-off across each condition. The x-axis shows the average time (second), and the y-axis shows the body mass normalised COM power (W/kg). Each colour coded line represents the average value across all participants for each condition. (B) Box and Whisker plot for body mass normalised concentric peak vertical COM power across non-AEL (BW_pre), 10%, 20%, and 30% AEL conditions. The x-axis shows different loading conditions, and the y-axis shows the body mass normalised vertical COM power (W/kg). (C) Box and Whisker plot for concentric vertical COM work across non-AEL (BW_pre), 10%, 20%, and 30% AEL conditions. The x-axis shows different loading conditions, and the y-axis shows the body mass normalised vertical COM work (J/kg). The asterisks represent a statistically significant difference from the mean of the baseline (BW_pre) condition.

0.086), knee (p = 0.002, $\eta^2_p$ = 0.143), and ankle (p = 0.033, $\eta^2_p$ = 0.056) joints. Overall, all three AEL conditions significantly increased the amount of negative work at the knee joint (10%: p = 0.002, d = 0.621; 20%: p = 0.001, d = 1.052; 30%: p = 0.020, d = 1.029), but significantly reduced the amount of negative work at the hip joint (10%: p = 0.001, d = 0.613; 20%: p = 0.006, d = 0.599; 30%: p = 0.004, d = 0.773). Multiple comparisons also showed that 30% AEL condition significantly increased the amount of negative work at the ankle joint compared to BW_pre condition (p = 0.038, d = 0.602).

## Vertical COM power and work

The average time-varying concentric vertical COM powers during push-off across each condition are shown in Fig 5A. There were no significant differences (p = 0.207, $\eta^2_p$ = 0.005) in the magnitude of the concentric peak vertical COM power between AEL conditions (Fig 5B). However, we found a significant main effect of AEL condition on the concentric vertical COM work (p = 0.028, $\eta^2_p$ = 0.024). Multiple comparison tests showed a significant reduction in the concentric vertical COM work in 20% (p = 0.022, d = 0.926) and 30% (p = 0.005, d = 0.914) AEL conditions compared to the baseline (BW_pre) (Fig 5C).

## Surface EMG

We found no significant differences between conditions in right VL or RF peak EMG amplitudes (VL: p = 0.546, $\eta^2_p$ = 0.036, RF: p = 0.234, $\eta^2_p$ = 0.055) or average integrated EMG (VL: p = 0.223, $\eta^2_p$ = 0.047; RF: p = 0.192, $\eta^2_p$ = 0.063) during push-off (Fig 6). We also found no significant differences in average peak EMG amplitudes (GLUT: p = 0.502, $\eta^2_p$ = 0.051, BF: p = 0.785, $\eta^2_p$ = 0.020, SOL: p = 0.757, $\eta^2_p$ = 0.020, MG: p = 0.020, $\eta^2_p$ = 0.172, S4 Fig) or average integrated EMG (GLUT: p = 0.086, $\eta^2_p$ = 0.110, BF: p = 0.151, $\eta^2_p$ = 0.045, SOL: p = 0.312, $\eta^2_p$ = 0.028, MG: p = 0.135, $\eta^2_p$ = 0.042, S5 Fig) between conditions in the other four lower limb muscles, except for the MG muscle which showed a significant reduction in peak EMG (main effect: p = 0.020, $\eta^2_p$ = 0.172) only in the 20% AEL condition (multiple comparison: p = 0.020, d = 0.957, S4 Fig).

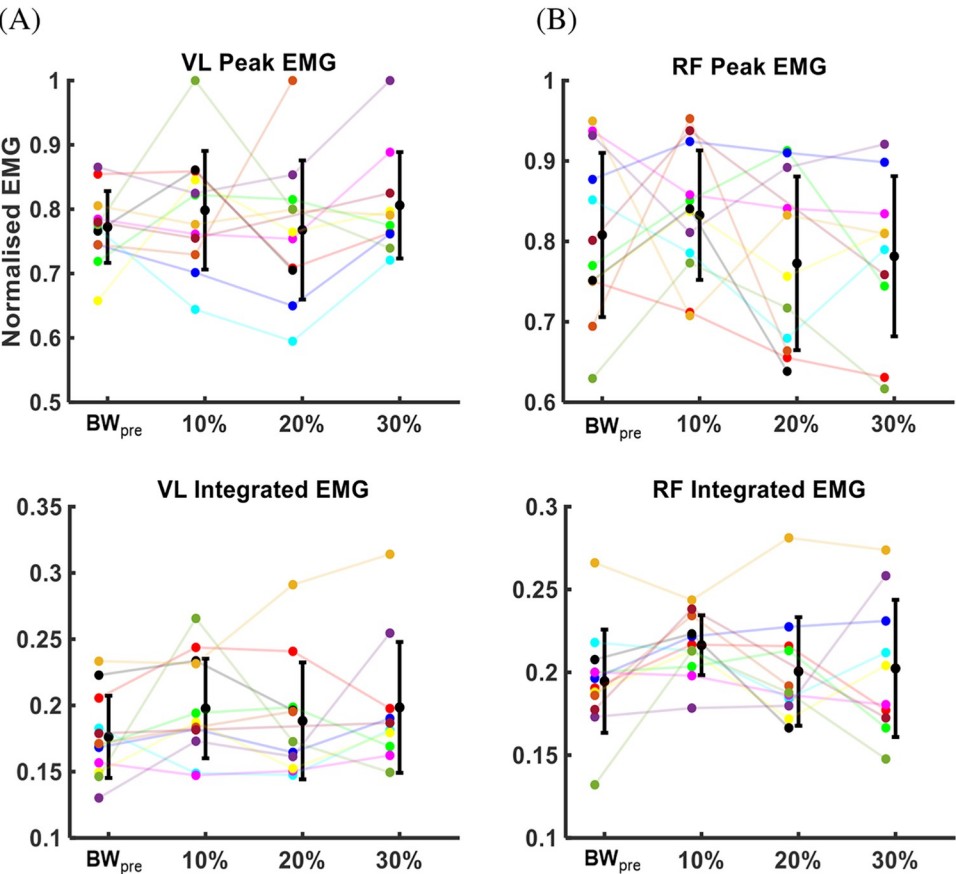

(A)

(B)

**Fig 6.** Mean and error bars for the average peak EMG and the average integrated EMG during the entire jump in (A) VL and (B) RF muscles. The x-axis shows different loading conditions, and the y-axis shows the normalised EMG. The smaller dots with different colour codes represent each participant's average peak or integrated EMG for each condition. The larger black dot represents the group mean, and the error bar represents one standard deviation from the group mean for each condition.

## Discussion

The results show that AEL did not result in performance enhancement in the constrained CMJ used in this study. We found no increase in vertical GRF at the start of the concentric phase nor increases in power or work in the concentric phase. Instead, we found slight decrements in performance in high load conditions. We therefore reject our main hypothesis that added weight alone increases the vertical ground reaction force (VGRF) at the start of the concentric movement and therefore increases the power and subsequent work during the concentric phase of a CMJ. We also reject our second hypothesis because we found no increase in the activation of lower limb muscles during the upward motion with AEL. Mechanistically, when other alternative movement factors were excluded, we found no enhancement in force applied or muscle activation at the start of the upward movement in any of the AEL CMJ conditions and this suggests that there is no additional storage and return of elastic energy during AEL CMJ from the lower limb muscles under the constrained conditions used.

Under the strictly constrained conditions imposed here, which controlled contributions from alternative factors that might affect human jump performance (i.e. squat depth, trunk and arm motion), the AEL intervention was found to decrease CMJ performance in the higher added mass conditons. For example, participants showed a decreased COM work during

push-off (Fig 5C) and decreased effective jump height (Fig 2A) for 20% and 30% AEL conditions compared to the baseline (BW$_{pre}$). This impaired performance during the AEL jump was associated with a small, but significant reduction in hip joint power (Fig 4A and 4B) and work during push-off, while other joints showed no change in work production. Given that the post AEL trial jump performance (BW$_{post}$) was unchanged at the group level compared to that before the AEL trials (BW$_{pre}$), the impairments in performance with AEL are unlikely to be a result of order effects or fatigue. Aboodarda et al. [1] and Sheppard et al. [2] reported that AEL improved CMJ jump height and other concentric kinetic and kinematic parameters (i.e., peak COM power, peak VGRF, and peak COM velocity). However, Aboodarda et al. [1] reported that participants increased squat depth to accumulate a longer contact time and larger impulse during push-off to improve jump height. While Sheppard et al. [2] did not observe any significant change in squat depth with AEL, neither the joint nor segment kinematics were reported. Potentially, participants in Sheppard et al. [2] might have selected different body configurations at the start of the push-off phase which helped to achieve superior performance. Changing body configuration, particularly the trunk angle at the bottom of the countermovement jump, could impact maximal jump performance [19]. Our study strictly controlled squat depth and body configuration at the bottom of the countermovement, which eliminated these possible effects during constrained AEL jumps.

Aboodarda et al. [1] reported an increased rate of eccentric loading and descent velocity during AEL, which could have contributed to performance enhancement via a more rapid stretch-shortening cycle. Previous research shows that applying a faster rate of muscle stretch improves muscle work and power more than a slower stretch [20,21]. In contrast, a study that involved performing barbell squats with a jump did not gain performance benefit by adding AEL during the eccentric phase of the squat [7]. Moore et al. [7] suggested that the relatively slow descent speed during the AEL barbell squat could have limited the performance enhancement effect of AEL. Our present study did not control the rate at which the descent phase was performed, and the slower descent speed observed during heavier AEL conditions might explain why our participants showed decreased performance compared to the non-AEL condition (Fig 2C). An alternative explanation could be that Aboodarda et al. [1] used elastic resistance instead of additional mass, which potentially changed the neuromuscular responses due to different loading mechanisms. On the other hand, the task of Moore et al. [7] required balancing a barbell and hence was more restrictive in trunk/hip flexion during the eccentric phase. As such, Moore's et al. [7] task required a movement pattern similar to our study, which may explain the similar findings (i.e. no AEL enhancement).

We believe our results refute the assumption that AEL performance effects can be attributed to enhanced ability to store and return energy with additional loading in the eccentric phase. In this study, squat depth and trunk rotation were purposefully constrained to examine differences in force output in a strictly controlled environment. Whilst the amount of mechanical work absorbed across all joints increased with increasing AEL weight, and VGRF was higher throughout most of the downward movement with increasing AEL weight (Fig 3A), there was no change in VGRF at the time of release of the weights in the AEL conditions compared to the body weight only condition. We also found no change in the ankle moment at the bottom of the movement across conditions, whilst knee joint moment increased with added weight (10.5% and 8.2% increase relative to the BW$_{pre}$ condition in the 20% and 30% AEL conditions respectively) and the hip joint moment decreased in by as much as 29% in the 30% AEL condition. Joint moments are indicative of muscle forces around joints, and therefore are directly related to energy stored in the in-series elastic tissues. Therefore the joint moment changes at the turning point of the jump with AEL suggests no change in elastic energy storage at the ankle (a key joint for storing and returning energy from the highly compliant Achilles tendon

[22]), a potential small increase in energy storage across the knee, and a reduction in energy storage potential across the hip (which likely has the lowest contribution of elastic energy storage and return). Whilst the knee had some increase in potential for storage of energy with added weight (considerably less than the increase in added load relative to body weight), this did not translate into greater power or work generation at the knee. We believe this is strong evidence for limited additional elastic energy storage and return capacity with AEL, since the added weight itself did not increase the VGRF at the time of release of the added weight, nor did it have substantial impact on increasing joint moments. However, since our study limited the hip and trunk motion, this may have reduced the capacity for additional loading at the knee and ankle, whilst also reducing the capacity for the hip to generate force and power that might be possible in unconstrained jumping. As such, a similar mechanical analysis to that performed here is required to gauge such effects in unconstrained jumping, as previous studies have not examined this movement with such metrics.

The results of this study support our earlier simulation study which found no performance enhancement in AEL for a simplified single-joint model [23]. Our present study showed that added mass did not increase the VGRF at the bottom of the countermovement despite a greater mean force during the descent (Fig 3A). Mechanically, this was achieved by generating force earlier in the descent and increasing the duration of the descent (Fig 3A), which inevitably slowed down the average descent speed in AEL conditions (Fig 2C). This movement characteristic was also predicted in our simulation study [23]. Our findings suggest that increased tendon-loading and elastic energy return is unlikely to contribute to the change in performance in a constrained human jumping motion, particularly when muscles are maximally activated at the bottom of the movement and producing a maximal achievable force. It is likely that other potential mechanisms, such as increasing squat depth, rate of descent or altered kinematics might be more relevant and contribute to enhanced performance.

There are some important limitations of our research that should be discussed. First, our study used a constrained jump in order to eliminate possible confounding effects during the experiment (i.e. change in squat depth or body configuration). While this was an important experimental control that added strength to our study, we could not directly compare our results to an unconstrained condition in which AEL had a performance-enhancing effect.

We selected a constrained single-joint task in this study to test if the proposed elastic energy return theory still holds true under such conditions. We examined the knee joint because it had the largest relative joint work contribution of a free CMJ (hip: 28%, knee: 49%, ankle: 23%) of all three lower limb joints [24]. However, such highly constrained movement pattern might disallow additional hip extensor work or storage of elastic energy in the Achilles tendon during the jump. These alternative sources of energy could have been critical in increasing jump heights during a natural free jumping, which were not allowed in this study. Despite this limitation, our understanding that jumping is a knee-dominant task provides a fundamental argument against attributing AEL effect purely to storing and returning elastic energy as a result of added load, and supports future research directions towards an efficient multi-joint coordination as the possible mechanism for AEL. Secondly, the statistical power of our study was set to detect changes measured during unconstrained jumps. Whilst it is possible that we were statistically underpowered given the changes in the constraints, the significant reduction in jump height and power measured with AEL would suggest that increasing participant numbers is unlikely to change the result. Another limitation was that we examined recreationally active adults, whereas previous studies typically used trained athletes with significant jumping experience and performance [1–4]. It is possible that untrained jumpers are not able to adapt to changes in mass as quickly as trained jumpers and could have been performing submaximally in the AEL conditions due to the novelty of the task. However, care was taken to ensure

jumpers had adequate training and familiarisation to the task, feedback on performance was given to try attaining maximal jump height, and we saw no statistical decrements in activations across AEL conditions (see S4–S6 Figs). We did observe inter-individual variability in the EMG response to different conditions (S4 and S5 Figs), and therefore we cannot rule out that some of our analysed jumps did not achieve maximum activation within the constraints of the movement task. Fluctuations in the tension in the cable with acceleration and deceleration of the added mass could have influenced the chosen motor pattern, however the accelerations were unlikely to have caused a substantial reductions in the force applied. Finally, although we used an objective method to release the mass at the beginning of the upwards movement, it is likely that there were small differences in the release timing across trials that might have contributed to the variability in the jump performances. In our experiment, the average release timing was 0.084 seconds after the turning point, with a small standard deviation of 0.05 seconds. In previous studies, participants were able to release weights at their own discretion [2,4], which probably also causes variation in release times relative to the movement. Future studies might specifically examine the effect of release timing on performance outcomes. It was also possible that participants in these studies [2,4] could have pushed off the released weights in order to gain extra upward impulse to further enhance jump performance. However, no data were available to confirm this speculation.

Our study provides an alternative view on how AEL might impact human lower limb work and power production during explosive movements. While AEL involves decelerating a heavier load in the eccentric phase, how the motion was performed was likely more critical than simply using a higher load mass. A higher elastic energy storage could only be achieved by a higher muscle force at the start of the push-off, whereas our study showed this was not always guaranteed with AEL. Our study could provide evidence against the effect of AEL for other similar movement configurations, such as for use in knee press machines or knee extension sleds of various angles. However, more research is required to inform coaching decision related to AEL implementation during natural, unconstrained jumps.

## Conclusion

In this study, AEL did not enhance performance during a trunk-constrained, knee-dominant maximal jump. We found that adding loads to the body did not change the knee extensor muscles' peak or integrated activation level, nor did it change the maximal VGRF at the start of the push-off. Therefore, AEL did not effectively increase force generation during the push-off phase after the load was released. Furthermore, AEL reduced the effective jump height in 20% and 30% AEL conditions, primarily due to a reduction in hip joint work. As such, we reject the premise that AEL increases jump performance due purely to increased muscle tension and hence storage of elastic energy. It is likely that other mechanisms related to rate of descent, squat depth, or body configuration changes in response to added weight contribute more to any potential AEL effects on jump performance.

## Supporting information

**S1 Fig. Joint moment data from the lower limb joints.** (A) Average time varying right hip, knee, and ankle joint moments across each condition. The x-axis shows the average time (second), and the y-axis shows the body mass normalised joint moment (N.m/kg) for hip, knee and ankle. Each colour coded line represents the average value across all participants for each condition. Time 0 represents the bottom of the countermovement (turning point). (B) Box and Whisker plots for normalised right hip extension, knee extension, and ankle plantar-flexion joint moments at the turning point across non-AEL (BW$_{pre}$), 10%, 20%, and 30% AEL

conditions. The x-axis shows different loading conditions, and the y-axis shows the normalised hip extension, knee extension, and ankle plantarflexion joint moments (N.m/kg) at the turning point. The asterisks represent a statistically significant difference from the mean of the baseline (BW$_{pre}$) condition.
(TIF)

**S2 Fig. Concentric joint work data from the lower limb joints.** Box and Whisker plots for right hip, knee, ankle, and sum of three joints concentric work (push-off) across non-AEL (BW$_{pre}$), 10%, 20%, and 30% AEL conditions. The x-axis shows different loading conditions, and the y-axis shows the body mass normalised joint work (J/kg). The asterisks represent a statistically significant difference from the mean of the baseline (BW$_{pre}$) condition.
(TIF)

**S3 Fig. Eccentric joint work data from the lower limb joints.** Box and Whisker plots for right hip, knee, and ankle joints eccentric/negative work across non-AEL (BW$_{pre}$), 10%, 20%, and 30% AEL conditions. The x-axis shows different loading conditions, and the y-axis shows the body mass normalised joint work (J/kg). The asterisks represent a statistically significant difference from the mean of the baseline (BW$_{pre}$) condition.
(TIF)

**S4 Fig. Normalised average peak EMG plots for all six lower limb muscles.** The x-axis shows different loading conditions, and the y-axis shows the normalised average peak EMG. We found no significant main effects in all muscles (mixed-effects model), except for MG muscle which showed a significant main effect. In post-hoc test, we found a significant reduction from baseline (BWpre) in the MG peak EMG in 20% AEL condition.
(TIF)

**S5 Fig. Normalised average integrated EMG plots for all six lower limb muscles.** The x-axis shows different loading conditions, and the y-axis shows the normalised average integrated EMG. We found no significant main effects between groups in all six muscles (mixed-effects model).
(TIF)

**S6 Fig. Average time varying normalised VL EMG during the entire jump across conditions.** The x-axis shows the average time (seconds), and the y-axis shows the normalised VL EMG. Each colour coded line represents the average value across all participants for each condition. Time 0 represents the bottom of the countermovement (turning point).
(TIF)

**S1 File.**
(ZIP)

## Acknowledgments

We are grateful to Prof. Andrew Cresswell (The University of Queensland) for assistance in developing custom Spike2 scripts in this study.

## Author Contributions

**Conceptualization:** Eric Yung-Sheng Su, Timothy J. Carroll, Dominic J. Farris, Glen Lichtwark.

**Formal analysis:** Eric Yung-Sheng Su.

**Funding acquisition:** Glen Lichtwark.

**Investigation:** Eric Yung-Sheng Su, Timothy J. Carroll, Dominic J. Farris, Glen Lichtwark.

**Methodology:** Eric Yung-Sheng Su, Timothy J. Carroll, Dominic J. Farris, Glen Lichtwark.

**Resources:** Glen Lichtwark.

**Supervision:** Timothy J. Carroll, Dominic J. Farris, Glen Lichtwark.

**Visualization:** Eric Yung-Sheng Su.

**Writing – original draft:** Eric Yung-Sheng Su.

**Writing – review & editing:** Eric Yung-Sheng Su, Timothy J. Carroll, Dominic J. Farris, Glen Lichtwark.

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
