## [Decision Letter · Decision Letter 0]

19 Mar 2024

PONE-D-24-04333Increased force and elastic energy storage are not the mechanisms that improve jump performance with accentuated eccentric loading during a constrained vertical jumpPLOS ONE

Dear Dr. Su,

Thank you for submitting your manuscript to PLOS ONE. After careful consideration, we feel that it has merit but does not fully meet PLOS ONE’s publication criteria as it currently stands. Therefore, we invite you to submit a revised version of the manuscript that addresses the points raised during the review process. Please review the comments attentively. Reviewer 1 has highlighted concerns that call into question the validity of the experiment. It is imperative that you address these points with detailed consideration. Please submit your revised manuscript by May 03 2024 11:59PM. If you will need more time than this to complete your revisions, please reply to this message or contact the journal office at plosone@plos.org. Please include the following items when submitting your revised manuscript:A rebuttal letter that responds to each point raised by the academic editor and reviewer(s). You should upload this letter as a separate file labeled 'Response to Reviewers'.A marked-up copy of your manuscript that highlights changes made to the original version. You should upload this as a separate file labeled 'Revised Manuscript with Track Changes'.An unmarked version of your revised paper without tracked changes. You should upload this as a separate file labeled 'Manuscript'.

We look forward to receiving your revised manuscript.

Kind regards,

Žiga Kozinc

Academic Editor

PLOS ONE

Additional Editor Comments:

Dear Authors,

Please review the comments attentively. Reviewer 1 has highlighted concerns that call into question the validity of the experiment. It is imperative that you address these points with detailed consideration.

Reviewers' comments:

Reviewer's Responses to Questions

**Comments to the Author**

1. Is the manuscript technically sound, and do the data support the conclusions?

Reviewer #1: Yes

Reviewer #2: Yes

Reviewer #3: Yes

2. Has the statistical analysis been performed appropriately and rigorously? 

Reviewer #1: Yes

Reviewer #2: Yes

Reviewer #3: Yes

3. Have the authors made all data underlying the findings in their manuscript fully available?

Reviewer #1: Yes

Reviewer #2: Yes

Reviewer #3: No

4. Is the manuscript presented in an intelligible fashion and written in standard English?

Reviewer #1: Yes

Reviewer #2: Yes

Reviewer #3: Yes

5. Review Comments to the Author

Reviewer #1: General Comments:

This study was to investigate the effect of additional eccentric loading on jump performance (jumping height and power). It was a very interesting topic and the entire study was well executed. In general, if the load in the ECC phase is higher (i.e., higher activity level), the jump height should increase because the load is lighter in the CON phase (since the additional load is removed in the CON phase in this study). Previous studies showed that optimal loading of the ECC phase existed with squat exercise (Takarada et al. 1997), drop-jump (Komi and Bosco 1978). However, in the present study, the work of hip joint and center of mass (and jumping height) decreased under high eccentric loading condition. This result may be related to the fact that the subjects were not giving their all due to the sudden load removal when switching from the eccentric phase to the concentric phase. In addition, there are similar concerns because jumping exercises using the device in Figure 1 are quite different from normal jumping exercises. If this is the case, I think it could lead to erroneous views in research on stretch-shortening cycle exercise. Throughout this manuscript, the additional loading of the eccentric phase is discussed in connection with the increase in elastic energy. However, this argument is disconcerting because the two are not necessarily the same.

Specific Comments:

Introduction

Line 42-44

In ref 9, the effects of AEL have not been investigated. This study was a cross-sectional comparison of tendon characteristics of athletes in different disciplines.

Line 62-64

As with the comment above, I disagree with this statement.

Reviewer #2: While seemingly simple, the execution of a countermovement jump is a complex sequence of generation and release of elastic energy. The current study is designed to determine if more elastic energy is released during the propulsive phase in response to enhancing only the eccentric load during the countermovement while controlling for all other variables. I commend the authors for designing an apparatus and methodology for controlling the many potentially confounding variables in the eccentric phase. However, this study may have been too constrained with too narrow a focus. For a complete understanding of the effects of AEL on jump performance, the entire force profile of the countermovement jump (i.e. both phases) should be examined. The study by McHugh et al. (Transl Sport Med, 2021) highlights the importance of 1) unweighting during the eccentric phase and 2) the timing of peak force to occur at the turning point, to jump performance and efficiency. While potential sources of variation such as trunk angle, and countermovement depth were controlled for in the current study, an explanation of the current findings may be found by examining eccentric phase metrics, instead of focusing only on the concentric phase. I recommend major revisions to this manuscript and the inclusion and comparison of eccentric phase force and joint metrics (see McHugh et al.) under the AEL conditions in order to completely characterize the effects of AEL on the whole countermovement jump profile.

Please see all individual comments below.

INTRODUCTION

The first paragraph of the introduction is well-written and succinctly summarizes the current research findings regrading AEL.

Lines 44-49: The amount of unweighting (defined by McHugh et al. as "low force") also plays an integral role in elastic energy storage and jump performance. While more force may be needed to decelerate during the eccentric braking phase under the AEL conditions, the amount of unweighting represents the "launch pad" from which to build tension and store elastic energy. With greater the unweighting (i.e. the lower the low force), a greater amount of tension can be accumulated and ultimately released. It is possible that AEL disrupts the mechanism of optimal unweighting, which may ultimately decrease performance. This is why the whole force curve must be examined, not only concentric phase metrics.

METHODS

The Methods are very detailed and precisely describe each aspect of the experiment and the equipment and set-up involved.

Line 195: Joint ranges of motion and a joint stiffness measurement may be more useful to characterize any changes occurring during the eccentric phase. Also, eccentric work should be considered as potential factor as there may be a optimal amount of energy which can be absorbed during the descent before an impairment in concentric energy generation occurs.

Lines 190-191: Why was average descent speed as a metric instead of maximal descent speed? The body must be decelerated from its maximum downward velocity by the end of the countermovement. The amount of force required to do this relative to the amount of unweighting represents the total amount of tension accumulated in the lower extremity. The average velocity throughout the whole countermovement does not accurately characterize the kinematics of the descent.

RESULTS

As stated earlier, the eccentric phase metric should be included in the analysis for complete characterization of the effects of AEL.

Figure 2C: I feel that maximum descent speed should be examined instead of average

Figure 3A: Based on the limited data presented in the graph, it does seem like there is a difference in the amount of unweighting between some of the conditions. It is difficult to determine this with certainty because the eccentric portion of the force profiles are not included. Please include the entire profile for each condition.

Also, it is clear that the peak concentric force does to coincide with the turning point in any of these conditions. According to McHugh et al., this is an indication of a biomechanically inefficient jump. This might be due to the constraining of so many of the eccentric phase variables using the sled. This point should be addressed in the discussion.

Figure 4A: Based on the profiles of the joint powers, it seems that there may be differences in the peak magnitude of eccentric power at the hip across conditions. However, this needs to be explored more rigorously. Additionally, the duration of the eccentric phase seems to increase in the AEL conditions, which may have implications on negative work done at each joint.

DISCUSSION

Line 322: Storage and return of elastic energy are dependent on the amount of unweighting and the total force generated during braking. As these variables were not included in the analysis, it is difficult to determine if changes to these variables were responsible for the negative effects on performance.

Reviewer #3: SUMMARY:

The study investigates the effect of accentuated eccentric loading or AEL, on the performance of human participants executing a countermovement jump. The authors used a sliding rail mechanism to control several aspects of the jumping movement in order to reduce extraneous factors not purely related to the added load during eccentric movements. Force plates, motion capture, and electromyography were used to measure ground reaction forces, joint and center of mass kinematics, and electrical activity of notable knee extensor muscles, respectively. Linear mixed-effects models were used to estimate the effect of eccentric load (10, 20 and 30% body weight) relative to a baseline jump with minimal eccentric load on multiple parameters including jump height, descent velocity, ground reaction forces, joint moments, power, and work. The study found a slight reduction in jumping height at higher eccentric loads, perhaps due to reduced hip extensor moments and power. The authors conclude that added eccentric load does not enhance jumping performance in a controlled jumping movement.

The study is rigorously designed and executed, and the manuscript is clearly written. In particular, I appreciate the authors’ honesty in rejecting both hypotheses which are reasonably developed based on previous research on the topic. In general, I believe this paper will add a nice contribution to the literature and will point researchers toward future questions more tightly aimed at understanding the AEL phenomenon and the factors underlying it. The following are a few minor-to-moderate comments which I hope can still improve the paper further.

REVIEWER COMMENTS:

Probably my biggest comment is related to the loading strategy used in the experiment. The authors suspend weight plates from a cable attached to the backrest where participants perform the jumping movement (either 10, 20 or 30% body weight). While this approach is mostly fine, there are some limitations to consider.

First, the static loading conditions are not actually static. That is, the tension force experienced by participants is closer to the load weight plus the inertial force of the load. If we assume that tension force is always greater than zero (see next point) and the cable is sufficiently rigid, then we can also assume that the vertical motion of the load and sliding rail are roughly equivalent during experiments. As such, any accelerations of the participants in the vertical direction also contribute to fluctuations in tension force. This could be one reason why descent speeds were slower at greater loads; that is, participants may have been avoiding greater tension fluctuations during descent by employing lower accelerations to reach lower descent speeds.

Second, if the participant ever accelerates downward at a rate higher than gravitational acceleration, then the tension in the cable can drop to zero. It is theoretically possible that individuals may have slowed their descent rate at loads greater than zero in order to prevent a zero tension eventually resulting in a large force spike once the weight caught tension again. This scenario is perhaps unlikely for multiple reasons, but since there is no measurement of tension force or acceleration during the eccentric motion, the reader cannot know this for sure.

Third, another form of tension fluctuation could have occurred due to centripetal accelerations of the weight plates swinging away from a vertical orientation. While the motion of the participants was fairly well-controlled, it is unclear whether the researchers made any attempt to control the motions of the weight plates during descent. This would perhaps have only a small effect, however.

In each of the above points, questions about the load being applied arise due to a missing measurement of the tension force during the experiment. While it would likely not be realistic to redo experiments with these measurements, a fairly accurate approximation of the load force could be calculated via the summation of inertial force and load weight: Tension=mg+my ¨ where y ¨ is vertical acceleration of the load (equal to vertical acceleration of the participant) and m is the load mass. Average tension during descent could be measured and even included in the statistical model as continuous data (rather than categorical) in order to potentially account for more of the data’s variability with loading condition. At the very least, it would be helpful to show an example of approximated tension from the calculation somewhere in the manuscript.

Line 119 – was friction in the linear rail system or pulleys assessed? Can the authors estimate how much additional load any friction or neglected inertial effects might have added? If not, it might be more transparent to mention that these effects were neglected in the analysis but are assumed to be quite low.

Line 122 – when the author refer to an “inextensible” metal wire, maybe they can report the gauge or diameter of the wire to help back up their claim of rigidity.

Lines 144, 177, 198 - in most cases, a number value is reported followed by a space and the units. However, there are a few instances where there is no space between the value and the units. Some examples are “9mm” (Line 144), “15Hz” (Line 177) and “5Hz” (Line 198). The authors should check the manuscript matches the journal formatting rules.

Line 128 – is it possible that the noise of the weights hitting the box could have affected participant performance? Perhaps this is part of a slightly larger discussion regarding participant instructions. For example, were participants told about what would happen to the weights during each jump? Also, how was “maximal effort” (Line 100) enforced or motivated?

Line 324 – it is interesting that the previous simulation study predicted some of the experimental results, and the discussion that follows in Lines 392-398 adds a new dimension to the interpretation. However, the initial mention of the simulation study in Line 324 feels out of place since no further context is given. I recommend waiting until the later discussion before referencing the simulation.

Line 395 – descent is misspelled here as “desent.”

Line 431-432 – I am not familiar with the studies referenced, but perhaps it is also possible that subjects in these studies could have pushed off the released weights in order to gain some amount of extra upward impulse.

The manuscript could be improved by adding either some discussion or background regarding the impact of the study and the topic more generally. Beyond developing a better understanding of the specific phenomenon related to human jumping, do the findings (or lack thereof) tell us something useful about training strategies or sports performance, for example?

6. PLOS authors have the option to publish the peer review history of their article (what does this mean?). If published, this will include your full peer review and any attached files.

Reviewer #1: No

Reviewer #2: No

Reviewer #3: No

---

## [Author Response · Author response to Decision Letter 0]

11 Jun 2024

Note: line numbers in our responses are based on the clean manuscript (not track changes).

Reviewer #1: General Comments:

This study was to investigate the effect of additional eccentric loading on jump performance (jumping height and power). It was a very interesting topic and the entire study was well executed. In general, if the load in the ECC phase is higher (i.e., higher activity level), the jump height should increase because the load is lighter in the CON phase (since the additional load is removed in the CON phase in this study). Previous studies showed that optimal loading of the ECC phase existed with squat exercise (Takarada et al. 1997), drop-jump (Komi and Bosco 1978). However, in the present study, the work of hip joint and center of mass (and jumping height) decreased under high eccentric loading condition. This result may be related to the fact that the subjects were not giving their all due to the sudden load removal when switching from the eccentric phase to the concentric phase. In addition, there are similar concerns because jumping exercises using the device in Figure 1 are quite different from normal jumping exercises. If this is the case, I think it could lead to erroneous views in research on stretch-shortening cycle exercise. Throughout this manuscript, the additional loading of the eccentric phase is discussed in connection with the increase in elastic energy. However, this argument is disconcerting because the two are not necessarily the same.

Response:

• Review Comment: “Subjects were not giving their all due to the sudden load removal when switching from the eccentric phase to the concentric phase.”

We understand the concern, however in our protocol we had familiarisation sessions and provided feedback to participants in all jumps to ensure they were ‘giving their all’. This is supported by our EMG data which showed no significant differences in average peak or average integrated EMG between different conditions in almost all muscles (please see S4-S5 Figs, and Line 380-387) As such we do not believe that participants gave less effort in the AEL conditions compared to the body weight only conditions. 

This is further evidenced in the VL EMG averaged time series data across the entire trial for all conditions (please see S6 Fig), which shows no evidence of a reduction in EMG level during the downward phase (in fact it is slightly increased with AEL conditions, although we have not statistically tested this), nor any changes in patterns due to the release in weight. Due to the rapid upward movement, it is unlikely that muscles can deactivate sufficiently quickly to change the performance, and there is no evidence in any of our EMG data that this occurs.

We have added a section to the discussion to address this concern – 

L519-526: “It is possible that untrained jumpers are not able to adapt to changes in mass as quickly as trained jumpers and could have performed submaximally in the AEL conditions due to the novelty of the task. However, care was taken to ensure jumpers had adequate training and familiarisation to the task, feedback on performance was given to try to attain maximal jump height, and we saw no decrements in activations across AEL conditions (see S4-S6 Figs). We expect that activation should be close to maximal for this type of task and hence an absence of EMG reductions at the time of release suggest that participants were not submaximal in the AEL conditions.”

• Review Comment: “In addition, there are similar concerns because jumping exercises using the device in Figure 1 are quite different from normal jumping exercises. If this is the case, I think it could lead to erroneous views in research on stretch-shortening cycle exercise.” 

While we accept that the constrained sled jumping used in not the same as a natural countermovement jump, we contend that this was entirely the point of the work - which aimed to isolate a potential mechanism for AEL rather than replicate prior work on the existence of AEL in jumping. That said, stretch-shortening cycle exercises exist in the form of movements as simple as a single-joint task (i.e., quick, cyclical elbow SSC curl) to movements as complicated as a countermovement jump. A jump is therefore a specific example of the general case of SSC’s, rather than vice-versa, and a more isolated SSC motion provides an opportunity to study the mechanisms underpinning that SSC’s more generally. Thus, using a modified movement task does not invalidate the scientific finding for the task in our study, particularly if our major focus was to examine whether increased tendon elastic energy storage and return could be observed in our SSC task. This is outlined in the introduction (L62-64) and then revisited in the Discussion (L407-411).

• Review Comment: “Throughout this manuscript, the additional loading of the eccentric phase is discussed in connection with the increase in elastic energy. However, this argument is disconcerting because the two are not necessarily the same.”

We understand that force and elastic energy are not the same - and we have been careful not to equate them in the manuscript - but they are intrinsically linked because energy stored in elastic tissues is proportional to the force applied. When the force is higher at the ground or at a joint, this implies greater energy storage in elastic tissues. We have made this clear in the introduction (L43-46) - 

“For instance, AEL CMJ may result in greater force generation in the descent to decelerate the added mass or resist the added force, potentially resulting in greater tendon loading and strain energy storage prior to the upward motion (strain of elastic tissues is directly related to force applied).”

We have also further clarified this in the discussion – 

L454-482: “Our results refute the assumption that AEL performance effects can be attributed to enhanced ability to store and return energy with additional loading in the eccentric phase. In this study, squat depth and trunk rotation were purposefully constrained to examine differences in force output in a strictly controlled environment. Whilst the amount of mechanical work absorbed across all joints increased with increasing AEL weight, and VGRF was higher throughout most of the downward movement with increasing AEL weight (Fig 3A), there was no change in VGRF at the time of release of the weights in the AEL conditions compared to the body weight only condition. We also found no change in the ankle moment at the bottom of the movement across conditions, whilst knee joint moment increased with added weight (10.5% and 8.2% increase relative to the BWpre condition in the 20% and 30% AEL conditions respectively) and the hip joint moment decreased by as much as 29% in the 30% AEL condition. Joint moments are indicative of muscle forces around joints, and therefore are directly related to energy stored in the in-series elastic tissues. Therefore the joint moment changes at the turning point of the jump with AEL suggests no change in elastic energy storage at the ankle (a key joint for storing and returning energy from the highly compliant Achilles tendon (22)), a potential small increase in energy storage across the knee, and a reduction in energy storage potential across the hip (which likely has the lowest contribution of elastic energy storage and return). Whilst the knee had some increase in potential for storage of energy with added weight (considerably less than the increase in added load relative to body weight), this did not translate into greater power or work generation at the knee. We believe this is strong evidence for limited additional elastic energy storage and return capacity with AEL, since the added weight itself did not increase the VGRF at the time of release of the added weight, nor have substantial impact on increasing joint moments. However, since our study limited the hip and trunk motion, this may have reduced the capacity for additional loading at the knee and ankle, whilst also reducing the capacity for the hip to generate force and power that might be possible in unconstrained jumping. As such, a similar mechanical analysis to that performed here is required to gauge such effects in unconstrained jumping, as previous studies have not examined this movement with such metrics.”

Specific Comments:

Introduction

Line 42-44

In ref 9, the effects of AEL have not been investigated. This study was a cross-sectional comparison of tendon characteristics of athletes in different disciplines.

Responses:

Thank you for your comment. We have now changed the reference to another paper that suggests elastic energy return improves AEL performance.

Line 62-64

As with the comment above, I disagree with this statement.

Responses:

It is not clear exactly which part of this statement the reviewer is disagreeing with based on the previous comments. However, as stated above, we contend that our experiment involves a stretch-shorten cycle with added load in the stretch. The fact the body’s movement is constrained (or at least the trunk movement) does not invalidate this, and in fact our design was such that we remove other factors that might also contribute to enhanced performance with AEL. As stated above, we have tried to make the introduction and discussion clearer. 

Reviewer #2: While seemingly simple, the execution of a countermovement jump is a complex sequence of generation and release of elastic energy. The current study is designed to determine if more elastic energy is released during the propulsive phase in response to enhancing only the eccentric load during the countermovement while controlling for all other variables. I commend the authors for designing an apparatus and methodology for controlling the many potentially confounding variables in the eccentric phase. However, this study may have been too constrained with too narrow a focus. For a complete understanding of the effects of AEL on jump performance, the entire force profile of the countermovement jump (i.e. both phases) should be examined. The study by McHugh et al. (Transl Sport Med, 2021) highlights the importance of 1) unweighting during the eccentric phase and 2) the timing of peak force to occur at the turning point, to jump performance and efficiency. While potential sources of variation such as trunk angle, and countermovement depth were controlled for in the current study, an explanation of the current findings may be found by examining eccentric phase metrics, instead of focusing only on the concentric phase. I recommend major revisions to this manuscript and the inclusion and comparison of eccentric phase force and joint metrics (see McHugh et al.) under the AEL conditions in order to completely characterize the effects of AEL on the whole countermovement jump profile.

Please see all individual comments below.

INTRODUCTION

The first paragraph of the introduction is well-written and succinctly summarizes the current research findings regrading AEL.

Lines 44-49: The amount of unweighting (defined by McHugh et al. as "low force") also plays an integral role in elastic energy storage and jump performance. While more force may be needed to decelerate during the eccentric braking phase under the AEL conditions, the amount of unweighting represents the "launch pad" from which to build tension and store elastic energy. With greater the unweighting (i.e. the lower the low force), a greater amount of tension can be accumulated and ultimately released. It is possible that AEL disrupts the mechanism of optimal unweighting, which may ultimately decrease performance. This is why the whole force curve must be examined, not only concentric phase metrics.

Responses:

Thank you for your valuable feedback and valid points about the importance of the eccentric phase in determining jump performance. We agree the force applied at the turning point depends on what occurs during the downward phase, but would like to reiterate comments made in responses to reviewer 1 that the active force is the primary factor that dictates elastic energy stored in the downward phase (force is directly related to stretch of elastic, tendinous tissues and hence energy stored). Energy stored in elastic tissues is not directly related to the work absorbed (negative work) during the downward movement, because some of this can be absorbed by muscle (ultimately output as heat). By design, energy absorbed (negative work) during the downward movement will be greater with added weight in the AEL conditions, because the gravitational force is higher and the displacement is the same. Our findings are important because the results clearly show that despite the increase in energy absorbed, there is no change in the force (either vertical ground reaction force or joint moments) at the time just before energy is returned, which means that energy stored due to the added weights is likely no different. Of course, as outlined in the discussion, this may be a function of the constraints placed on the movement, but the added mass alone is insufficient to increase the force at turning point when mass is released in the AEL conditions compared to the non-AEL condition. 

We have reworded a key paragraph in the discussion to make this clearer–

L454-482: “Our results refute the assumption that AEL performance effects can be attributed to enhanced ability to store and return energy with additional loading in the eccentric phase. In this study, squat depth and trunk rotation were purposefully constrained to examine differences in force output in a strictly controlled environment. Whilst the amount of mechanical work absorbed across all joints increased with increasing AEL weight, and VGRF was higher throughout most of the downward movement with increasing AEL weight (Fig 3A), there was no change in VGRF at the time of release of the weights in the AEL conditions compared to the body weight only condition. We also found no change in the ankle moment at the bottom of the movement across conditions, whilst knee joint moment increased with added weight (10.5% and 8.2% increase relative to the BWpre condition in the 20% and 30% AEL conditions respectively) and the hip joint moment decreased by as much as 29% in the 30% AEL condition. Joint moments are indicative of muscle forces around joints, and therefore are directly related to energy stored in the in-series elastic tissues. Therefore the joint moment changes at the turning point of the jump with AEL suggests no change in elastic energy storage at the ankle (a key joint for storing and returning energy from the highly compliant Achilles tendon (22)), a potential small increase in energy storage across the knee, and a reduction in energy storage potential across the hip (which likely has the lowest contribution of elastic energy storage and return). Whilst the knee had some increase in potential for storage of energy with added weight (considerably less than the increase in added load relative to body weight), this did not translate into greater power or work generation at the knee. We believe this is strong evidence for limited additional elastic energy storage and return capacity with AEL, since the added weight itself did not increase the VGRF at the time of release of the added weight, nor have substantial impact on increasing joint moments. However, since our study limited the hip and trunk motion, this may have reduced the capacity for additional loading at the knee and ankle, whilst also reducing the capacity for the hip to generate force and power that might be possible in unconstrained jumping. As such, a similar mechanical analysis to that performed here is required to gauge such effects in unconstrained jumping, as previous studies have not examined this movement with such metrics.”

We agree that the AEL likely disrupts the unweighting, with a different force profile required to first decelerate the combined masses and then accelerate upwards, however we are not 100% sure what metrics in the downward phase would be best examined to understand why AEL does not enhance performance in our setup. By definition, the VGRF (VGRF=ma+mg) must be higher throughout due to the added mass. As stated, work absorbed must also be higher if we have the 

---

## [Decision Letter · Decision Letter 1]

9 Jul 2024

PONE-D-24-04333R1Increased force and elastic energy storage are not the mechanisms that improve jump performance with accentuated eccentric loading during a constrained vertical jumpPLOS ONE

Dear Dr. Su,

Thank you for submitting your manuscript to PLOS ONE. After careful consideration, we feel that it has merit but does not fully meet PLOS ONE’s publication criteria as it currently stands. Therefore, we invite you to submit a revised version of the manuscript that addresses the points raised during the review process.

We look forward to receiving your revised manuscript.

Kind regards,

Žiga Kozinc

Academic Editor

PLOS ONE

Additional Editor Comments:

Dear Authors,

thank you for all the efforts in revising your manuscript. Reviewer 1 responded again, stating they are not completely satisfied with your responses. Please see their comments and try to see if you can be even more critical towards your results (eg., see the point on inter-individual variability in EMG responses) After this final revision, we will make a final decision.

Kind regards,

Reviewers' comments:

Reviewer's Responses to Questions

**Comments to the Author**

1. If the authors have adequately addressed your comments raised in a previous round of review and you feel that this manuscript is now acceptable for publication, you may indicate that here to bypass the “Comments to the Author” section, enter your conflict of interest statement in the “Confidential to Editor” section, and submit your "Accept" recommendation.

Reviewer #1: All comments have been addressed

Reviewer #2: All comments have been addressed

Reviewer #3: All comments have been addressed

2. Is the manuscript technically sound, and do the data support the conclusions?

Reviewer #1: Partly

Reviewer #2: Yes

Reviewer #3: Yes

3. Has the statistical analysis been performed appropriately and rigorously? 

Reviewer #1: Yes

Reviewer #2: Yes

Reviewer #3: Yes

4. Have the authors made all data underlying the findings in their manuscript fully available?

Reviewer #1: Yes

Reviewer #2: Yes

Reviewer #3: Yes

5. Is the manuscript presented in an intelligible fashion and written in standard English?

Reviewer #1: Yes

Reviewer #2: Yes

Reviewer #3: Yes

6. Review Comments to the Author

Reviewer #1: Thank you for your responses. As pointed out previously, however, I disagreed with two points.

Firstly, the reduction of the works of hip joint and center of mass with higher eccentric loading conditions may be related to the fact that the participants were not giving their all due to the sudden load removal when switching from the eccentric phase to the concentric phase. The authors rejected this possibility, because no differences in EMG data among all conditions were found. According to the individual EMG data (Figure 6), it is recognized that there were significant individual differences in the changes in EMG data in response to loading conditions. The individual differences were particularly pronounced in the GLUT (Fig. S4) and BF (Fig. S5). Because these muscle groups were closely related to hip joint work, I do not consider it possible to fully explain why the points I have identified (see above) are not involved in the results of this study (lower hip work with heavy loading).

Secondly, the additional loading of the eccentric phase is discussed in connection with the increase in elastic energy in this study. However, this argument is disconcerting because the two are not necessarily the same. Previous studies showed that optimal loading of the ECC phase existed with squat exercise (Takarada et al. 1997), drop-jump (Komi and Bosco 1978). For example, Takarada et al (1997) reported that power output during the concentric phase increased initially with the eccentric force, whereas they began to decline when the eccentric force exceeded >1.4 times the sum of load and body weight (JAP 83: 1749-1755).

Reviewer #2: The authors have thoroughly addressed my comments raised in a previous round of review. I feel that this manuscript is now acceptable for publication. Well done!

Reviewer #3: In general, I am satisfied by the author responses to the previous round of comments. I have listed a few minor editorial comments for the authors to consider, but otherwise I am happy to recommend publication.

Note, the following line numbers refer to the tracked-changes version of the manuscript.

Line 100: the way this line is written could be misunderstood by the reader to mean that the additional mass is spatially configured on top of the backrest. Perhaps this line could be updated for clarity: “…other AEL conditions included extra mass in addition to the mass of the backrest.”

Line 198: The letter “s” in OpenSim should be uppercase.

Line 483: change to “…the added weight, nor did it have…”

Line 528: the word novelty is misspelled as “novely”

Line 535-536: it is unclear what qualifies as a “significant reduction in force applied”, since I do not believe the authors are referring to statistical significance here (if they are, the statistics should be reported, e.g., p-value, etc.). Otherwise, I suggest a slight rewording, e.g., “…however the accelerations were unlikely to have caused a substantial reduction in the force applied.” Furthermore, this statement could be strengthened by quantification, e.g., “accelerations caused an average fluctuation in the force applied of X % participant body weight, and so are unlikely to have had a large influence on the motor patterns of participants.”

Line 554: Configuration should be plural. Similarly, knee press machine and knee extension sled should be plural, or else they should be preceded by the word “a”, e.g., “such as for use in a knee press machine…”

7. PLOS authors have the option to publish the peer review history of their article (what does this mean?). If published, this will include your full peer review and any attached files.

Reviewer #1: No

Reviewer #2: No

Reviewer #3: No

---

## [Author Response · Author response to Decision Letter 1]

17 Jul 2024

Reviewer #1: Thank you for your responses. As pointed out previously, however, I disagreed with two points.

Firstly, the reduction of the works of hip joint and center of mass with higher eccentric loading conditions may be related to the fact that the participants were not giving their all due to the sudden load removal when switching from the eccentric phase to the concentric phase. The authors rejected this possibility, because no differences in EMG data among all conditions were found. According to the individual EMG data (Figure 6), it is recognized that there were significant individual differences in the changes in EMG data in response to loading conditions. The individual differences were particularly pronounced in the GLUT (Fig. S4) and BF (Fig. S5). Because these muscle groups were closely related to hip joint work, I do not consider it possible to fully explain why the points I have identified (see above) are not involved in the results of this study (lower hip work with heavy loading).

Secondly, the additional loading of the eccentric phase is discussed in connection with the increase in elastic energy in this study. However, this argument is disconcerting because the two are not necessarily the same. Previous studies showed that optimal loading of the ECC phase existed with squat exercise (Takarada et al. 1997), drop-jump (Komi and Bosco 1978). For example, Takarada et al (1997) reported that power output during the concentric phase increased initially with the eccentric force, whereas they began to decline when the eccentric force exceeded >1.4 times the sum of load and body weight (JAP 83: 1749-1755).

Response:

Concern with submaximal effort with AEL - 

After carefully reviewing EMG data in Figure 6, Figure S4 and Figure S5, we agree that our EMG data showed some inter-individual variability in the EMG responses in muscles across the hip joint (GLUT and BF). However, single muscle EMG response can be highly variable across individuals for multi-joint dynamic movements (i.e., jumps), and seeing individual variability in EMG response does not necessarily mean submaximal voluntary effort in the jumps. 

We have acknowledged the possibility that participants may not use maximal effort during AEL jumps in the discussion of the manuscript, but this is hard to test in our study based on the available EMG data. As our EMG data did not show any statistically significant difference between conditions for GLUT and BF muscles, our overall data do not generally support this assertion. We have addressed this concern in the manuscript (L522-L527).

L522-L527: “However, care was taken to ensure jumpers had adequate training and familiarisation to the task, feedback on performance was given to try attaining maximal jump height, and we saw no statistical decrements in activations across AEL conditions (see S4-S6 Figs). We did observe inter-individual variability in the EMG response to different conditions (S4-S5 Figs), and therefore we cannot rule out that some of our analysed jumps did not achieve maximum activation within the constraints of the movement task.”

L527-L529: We removed this sentence in our manuscript -“We expect that activation should be close to maximal for this type of task and hence no reductions at the timing of release suggest that participants were not submaximal in the AEL conditions.”

Concern with “additional loading of the eccentric phase is discussed in connection with the increase in elastic energy in this study” - 

We believe that the reviewer is making the argument that elastic energy storage is related to the absorption of energy during the eccentric phase, and not to the load experienced and that as such we ‘should’ find that energy return is higher unless our design or constraints prevent maximum effort. The reviewer mentioned two studies (Takarada et al. 1997; Komi and Bosco 1978) using a range of loads to determine “optimal load” for the movements being tested, but these studies assessed power output with different loads in the concentric phase. Whilst we understand the reviewer’s point of view, we think there are a number of differences between our study and those cited, which make the comparison unfair. 

The cited studies (Takarada et al. 1997; Komi and Bosco 1978) used a fixed load during uncontrolled jumps - these were non-AEL tasks. The typical power-load relationship during squat exercise (Takarada et al. 1997) was the combined effect of both the concentric muscle force and the concentric movement velocity, with the latter being heavily constrained by the additional load in the body-mass system when the load mass exceeded a certain limit (1.4 times body + load mass); higher mass slows the concentric phase allowing muscles to generate more power according to the velocity-power relationship. However, this well-established finding cannot explain our AEL jump study because our concentric load was always the same load (relatively light load), regardless of the conditions. In our experiment, we also ensured we controlled squat depth (with the velocity of the squat also being matched, although not specifically controlled). Whilst we see from the Takarada et al. (1997) study that there was an increase in power with weight up to 1.4 times body + load mass, and that this was related (although not statistically tested) to a small increase in force prior to the upward movement, this experiment did not control squat depth. Similarly, the Komi and Bosco (1978) study also did not control squat depth and the input energy from changing drop jump height cannot be considered equivalent to our study. Furthermore, the relationship between negative work and elastic energy is not straight forward, given that ultrasound studies have clearly shown that muscles (particularly at the knee) actively absorb some of the energy by lengthening rather than storing all of this energy as was assumed in the Komi and Bosco (1978) paper (see for example Nikolaidou et al, Royal Society Open Science, 2017). 

When we calculate the mechanical energy absorbed in the eccentric phase, we see that this increases with increased load; this must be the case as we controlled squat depth (greater mass, same displacement = great change in potential energy). However, this is clearly not related to increased jump height in our results as we do not see any increase in jump height. The primary aim of our study was to objectively assess whether the VGRF at the turning point was also higher with AEL, given a higher mean VGRF during the eccentric phase. We then used this data to evaluate whether this relates to greater power/work in the upwards phase. We believe we have made a sound argument to link the forces at the instant at the turning point to potential elastic energy storage (L43-46) and justify these interpretations further in detail in the discussion, with the added use of the word ‘We believe’ to ensure the reader understands that the interpretation is our own and not necessarily proven (see L454-L482): 

“We believe our results refute the assumption that AEL performance effects can be attributed to enhanced ability to store and return energy with additional loading in the eccentric phase. In this study, squat depth and trunk rotation were purposefully constrained to examine differences in force output in a strictly controlled environment. Whilst the amount of mechanical work absorbed across all joints increased with increasing AEL weight, and VGRF was higher throughout most of the downward movement with increasing AEL weight (Fig 3A), there was no change in VGRF at the time of release of the weights in the AEL conditions compared to the body weight only condition. We also found no change in the ankle moment at the bottom of the movement across conditions, whilst knee joint moment increased with added weight (10.5% and 8.2% increase relative to the BWpre condition in the 20% and 30% AEL conditions respectively) and the hip joint moment decreased in by as much as 29% in the 30% AEL condition. Joint moments are indicative of muscle forces around joints, and therefore are directly related to energy stored in the in-series elastic tissues. Therefore the joint moment changes at the turning point of the jump with AEL suggests no change in elastic energy storage at the ankle (a key joint for storing and returning energy from the highly compliant Achilles tendon (22)), a potential small increase in energy storage across the knee, and a reduction in energy storage potential across the hip (which likely has the lowest contribution of elastic energy storage and return). Whilst the knee had some increase in potential for storage of energy with added weight (considerably less than the increase in added load relative to body weight), this did not translate into greater power or work generation at the knee. We believe this is strong evidence for limited additional elastic energy storage and return capacity with AEL, since the added weight itself did not increase the VGRF at the time of release of the added weight, nor have substantial impact on increasing joint moments. However, since our study limited the hip and trunk motion, this may have reduced the capacity for additional loading at the knee and ankle, whilst also reducing the capacity for the hip to generate force and power that might be possible in unconstrained jumping. As such, a similar mechanical analysis to that performed here is required to gauge such effects in unconstrained jumping, as previous studies have not examined this movement with such metrics.”

We have addressed numerous limitations or cautions in interpretations (including around maximal effort) in the discussion to ensure that the reviewer’s concerns are also considered alongside our interpretation. 

Reviewer #2: The authors have thoroughly addressed my comments raised in a previous round of review. I feel that this manuscript is now acceptable for publication. Well done!

Reviewer #3: In general, I am satisfied by the author responses to the previous round of comments. I have listed a few minor editorial comments for the authors to consider, but otherwise I am happy to recommend publication.

Note, the following line numbers refer to the tracked-changes version of the manuscript.

Line 100: the way this line is written could be misunderstood by the reader to mean that the additional mass is spatially configured on top of the backrest. Perhaps this line could be updated for clarity: “…other AEL conditions included extra mass in addition to the mass of the backrest.”

Response: We have updated the sentence as requested (see L99-L100).

Line 198: The letter “s” in OpenSim should be uppercase.

Response: We have changed the letter “s” to uppercase for all OpenSim words in the manuscript that were in lowercase (see L192 and L197).

Line 483: change to “…the added weight, nor did it have…”

Response: We have updated the sentence as requested (see L476).

Line 528: the word novelty is misspelled as “novely”

Response: We have corrected the spelling as requested (see L521).

Line 535-536: it is unclear what qualifies as a “significant reduction in force applied”, since I do not believe the authors are referring to statistical significance here (if they are, the statistics should be reported, e.g., p-value, etc.). Otherwise, I suggest a slight rewording, e.g., “…however the accelerations were unlikely to have caused a substantial reduction in the force applied.” Furthermore, this statement could be strengthened by quantification, e.g., “accelerations caused an average fluctuation in the force applied of X % participant body weight, and so are unlikely to have had a large influence on the motor patterns of participants.”

Response: We have changed the sentence to “…however the accelerations were unlikely to have caused a substantial reduction in the force applied.” (L531-532).

Line 554: Configuration should be plural. Similarly, knee press machine and knee extension sled should be plural, or else they should be preceded by the word “a”, e.g., “such as for use in a knee press machine…”

Response: We have changed these words to plural (i.e., configurations, machines, sleds) in L550-L551.

---

## [Editor Report · Decision Letter 2]

19 Jul 2024

Increased force and elastic energy storage are not the mechanisms that improve jump performance with accentuated eccentric loading during a constrained vertical jump

PONE-D-24-04333R2

Dear Dr. Su,

We’re pleased to inform you that your manuscript has been judged scientifically suitable for publication and will be formally accepted for publication once it meets all outstanding technical requirements.

Kind regards,

Žiga Kozinc

Academic Editor

PLOS ONE

Additional Editor Comments (optional):

Thank you for further amending the manuscript and putting more emphasis on concerns raised by Reviewer 1. I will recommend the acceptance of the manuscript.
---

## [Editor Report · Acceptance letter]

25 Jul 2024

PONE-D-24-04333R2 

PLOS ONE

Dear Dr. Su, 

I'm pleased to inform you that your manuscript has been deemed suitable for publication in PLOS ONE. Congratulations! Your manuscript is now being handed over to our production team.

Kind regards, 

on behalf of

Dr. Žiga Kozinc 

Academic Editor

PLOS ONE